



# Parameterisation toolbox for physical-biogeochemical model compatible with FABM. Case study: the coupled 1D GOTM-ECOSMO E2E for the Sylt-Romo Bight, North Sea

Hoa Nguyen[1,3], Ute Daewel[1], Neil Banas[2], and Corinna Schrum[1]

[1]Helmholtz-Zentrum Hereon, Institute of Coastal Research, Max-Planck-Str. 1, 21502 Geesthacht, Germany
[2]Department of Mathematics & Statistics, University of Strathclyde, 26 Richmond St., Glasgow, G1 1XH, UK
[3]Institute for Environment and Resources, Vietnam National University - HoChiMinh City, Vietnam

**Correspondence:** Hoa Nguyen (hoa.nguyen@hereon.de)

**Abstract.** Mathematical models serve as invaluable tool for comprehending marine ecosystems. The performance of these models is often highly dependent on their parameters. Traditionally, refining these models involved a time-intensive trial-and-error approach to identify model parameter values that are able to reproduce observations well. However, as ecosystem models grow in complexity, this approach becomes impractical. With advancements in computing power, optimization techniques have emerged as a viable alternative. Yet, these techniques often exhibit model-specific tailoring, limiting their broader application. In this study, we introduce a parameterisation toolbox founded on a Particle Swarm Optimizer (PSO) implemented in the Framework for Aquatic Biogeochemical Models (FABM), which allows its reuse between numerous existing models in FABM, and thus makes the optimizer more accessible to the community. The PSO toolbox's effectiveness is demonstrated through its implementation on a 1D physical-biogeochemical model (GOTM-ECOSMO E2E), which successfully parameterised the Sylt-Romo Bight ecosystem. The toolbox was able to identify most of the tuned parameters and to suggest potential ranges for poorly constrained parameters. In addition, the toolbox uncovers a number of parameter sets with notable differences in some parameter values, but resulting in not much difference in biomass and fluxes. Furthermore, by experimenting with optimisation models of varying complexity, the toolbox was able to define an optimal model for the Sylt-Romo Bight.

## 1 Introduction

Marine biogeochemical models serve as a valuable tools for hypothesis generation and gaining mechanistic understanding of ecosystem functioning. This understanding is particularly important in recent years, given the impact of anthropogenic-driven climate change and the increasing need to implement policies and technologies for effective mitigation. To achieve a comprehensive understanding of ecosystem functioning, it is essential to use models that are capable of reproducing the processes relevant to the system as closely as possible to observed data and real-world phenomena. However, obtaining such optimized models is challenging due to the complexity of the marine ecosystem. Firstly, unlike atmospheric and hydrodynamic models grounded in well-established physical laws such as the Navier-Stokes equations, the governing equations for marine biogeochemical models remain incomplete and have not been devised (Fennel et al., 2001; Jones et al., 2010; Schartau et al.,





2017). Secondly, the sparse nature of available data often hinders the constraint of all model parameters (Schartau et al., 2017). Thirdly, to cover the full extent of the marine ecosystem, a number of simplifications (e.g. the use of plankton functional

types) are required to reduce complexity and computational effort. As a result, processes in marine biogeochemical models rely heavily on parameterisation, which in the best case is often based on empirical studies (Miller, 2009). However, laboratory experiments to define parameters are typically conducted on a single species under controlled conditions, raising questions about their applicability to *in situ* conditions (Fennel et al., 2001). As an alternative approach, parameter optimization has been proposed as a means to address these challenges (Fennel et al., 2001; Dowd, 2011).

Parameter optimization has been used in marine ecosystem modelling to optimize poorly known model parameters (Prieß et al., 2013; Falls et al., 2022; Kern et al., 2024). Essentially, the optimization is done by fitting the model outputs to observed data by subjective tuning of the parameters. The parameters are varied until the misfit between model outputs and the observed data, often termed the cost function, is minimized (Fennel et al., 2001). A variety of optimization methods used in marine ecosystem modeling (e.g., Generalized Likelihood Uncertainty Estimation (GLUE, Beven and Binley (1992), Simulated An-

nealing (SA, Kirkpatrick et al. (1983), Markov Chain Monte Carlo Metropolis et al. (1953)) have been summarized in Houska (2017). Numerous studies have succeeded in implementation parameterisation and its advanced method in marine ecosystem modelling (e.g., (Rückelt et al., 2009; Prieß et al., 2011; Reimer, 2019)). However, these implementation are often tailored to specific models or difficult to reuse, requiring a lot of effort to implement a new parameterisation (Hemmings et al., 2015).

In biogeochemical modelling, the Framework for Aquatic Biogeochemical Models (FABM) is widely used to couple sev-

eral physical and ecological models. Many important hydrodynamic and biogeochemical models are available in FABM (e.g, hydrodynamic models include GOTM, ROMS, NEMO, MOM, HYCOM, FVCOM and SCHISM; ecosystem models include ECOSMO, ERSEM, BFM, PISCES (marine), and WET/PCLake (freshwater)). An optimisation method that is compatible with FABM would enable its use and promote its practice within the community. Therefore, in this paper, we present a parameterisation toolbox that is compatible with FABM and also to introduce a optimisation algorithm (Particle Swarm Optimizer

(PSO)) not often applied in biogeochemical models.

We demonstrate the application of the PSO to 1D GOTM-ECOSMO/ECOSMO E2E in FABM to optimise the model for the Sylt-Romo Bight ecosystem (hereafter the Sylt ecosystem) using observational data at the Sylt Road. We choose the Sylt ecosytem as an application because there are several hypotheses about the dynamics of the Sylt ecosystem, such as: (i) warmer winters lead to shifts from a pelagic to a more benthic dominated food web, (ii) enhanced top-down control during warm

winters by predators (e.g., blue mussels, oysters, razor clams) (*pers.comm. Johannes Rick and Sabine Horn*), (iii) bare mussel beds and mussel cultures reduce the standing stock of phytoplankton, but also promote phytoplankton primary production (Asmus and Asmus, 1991), to which requires a modelling approach to address.

The structure of the paper is as follows: in the next section we describe the PSO, the cost function used in the optimisation, the coupled 1D GOTM-ECOSMO and ECOSMO E2E (hereafter E2E) model configuration and setup. In the third section, we

present the results and discussion of the PSO for the Sylt ecosystem . Finally we end the paper with some concluding remarks.




## 2   Data and Methods

### 2.1   Parameterisation toolbox

The core of the paper is the implementation of the Particle Swarm Optimiser (PSO) with FABM as the PSO Toolbox. The PSO algorithm has been well described by Poli et al. (2007) and consequently well reviewed by Garcia-Gonzalo and Fernandez-Martinez (2012); Sengupta et al. (2019). The PSO from Poli et al. (2007) with a bound condition adjustment was used to identify parameters of an ecosystem model in Puget Sound (Nguyen, 2021). The PSO toolbox presented in this paper is based on the algorithm described in (Nguyen, 2021). A brief description of the PSO in plain language is given below; a full description can be found in (Nguyen, 2021).

The PSO algorithm can easily be explained by analogy with a process that uses boats to measure the deepest part of a large lake. First, imagine a large lake whose depth needs to be measured. It is almost impossible to do this with one boat. A more reasonable approach is to use several boats, and importantly, these boats need to communicate with each other about their measurements. To complete the above task, the first step is to randomly position two boats, A and B, on opposite sides of the lake. They both then measure the depths at their first positions, record them as their personal (A's and B's) deepest, and inform each other of their measurements. If A's first measurement is deeper than B's, then A's measurement is recorded as the global deepest. Next, in the second step, A stays where it is, as A has the global deepest, while B moves towards A by a predefined distance to a new position in the lake. B again measures the depth of the new position. B then compares the new depth with its personal deepest, and updates the personal deepest if the new depth appears to be deeper. B then exchanges its updated personal deepest with A and updates the global deepest if B's personal deepest is now deeper. Then the one with the deepest stays and the other moves. This process is repeated until A and B meet (converge) in the same place (the deepest point in the lake). The boats form a swarm, where each boat is a particle of the swarm. The deepest position represents the cost function. Obviously, it takes fewer iterations (less time) to find the deepest position if more boats are used.

**Algorithm.** Mathematically, the algorithm is presented as in figure 1. A particle i of the swarm at time t is characterised by its vector position $\overrightarrow{X_i}(t)$, vector velocity $\overrightarrow{vi}(t)$, and its personal cost $\overrightarrow{P_i}(t)$. The swarm at time t records its best (global) cost $\overrightarrow{G}(t)$. The movement of particle i from time t to time (t+1) with velocity $\overrightarrow{V_i}(t+1)$ must take into account its current vector velocity, its personal cost and the global (swarm) cost to reach position $\overrightarrow{X_i}(t+1)$ that is closer to the swarm's best position. Thus, the particle i moves first parallel to its current velocity vector $(\overrightarrow{vi}(t))$, then parallel to the vector connecting its current position $(\overrightarrow{X_i}(t))$ to its personal best position $(\overrightarrow{P_i}(t))$, and finally parallel to the vector connecting the current position $(\overrightarrow{X_i}(t))$ to the global best $(\overrightarrow{G}(t))$. The addition of these three vectors from the beginning of the first vector to the end of the third vector is its new velocity $(\overrightarrow{V_i}(t+1))$. Since new position of particle i is determined using the previous experience of the particle itself and of the whole swarm, the new position is considered to be the better position for particle i to be. If each particle in the swarm follows these rules, they will cooperate to find the best position in the search space, and thus the best possible solution. The algorithm is implemented as follows.



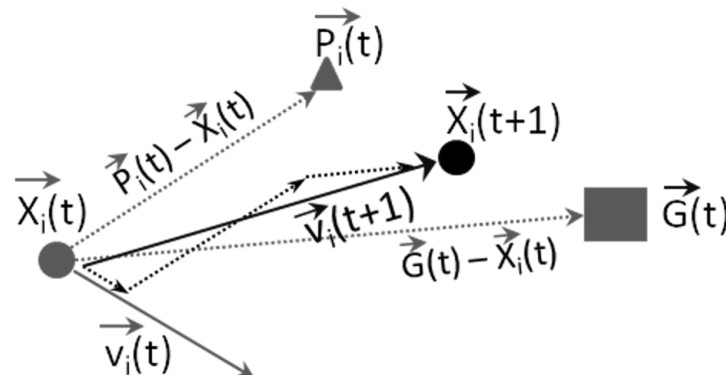

**Figure 1.** Particle Swarm Optimization Algorithm. A particle i of the swarm at time t is characterised by its vector position $\overrightarrow{X_i}(t)$, its vector velocity $\overrightarrow{vi}(t)$, and its personal cost $\overrightarrow{P_i}(t)$. The swarm at time t records its best (global) cost $\overrightarrow{G}(t)$. The particle i approaches to the global best by first moving parallel to its current velocity vector ($\overrightarrow{vi}(t)$), then parallel to the vector connecting current position ($\overrightarrow{X_i}(t)$) to its personal best ($\overrightarrow{P_i}(t)$), and finally parallel to the vector connecting the current position ($\overrightarrow{X_i}(t)$) to the global best ($\overrightarrow{G}(t)$). The addition of these three vectors from the beginning of the first vector to the end of the third vector is its new velocity ($\overrightarrow{V_i}(t+1)$).

1. Initialise a population array of particles with random positions and velocities on D dimensions in the search space rescaled to the interval (0, 1). The personal costs of the particles, calculated from initialised positions and velocities, are assigned to their $pbest$.

2. **loop**

3. For each particle, evaluate the desired optimisation fitness function in D variables.

4. Compare the fitness evaluation of the particle with its $pbest_i$. If the current value is better than $pbest_i$, then set $pbest_i$ equal to the current value, and $\boldsymbol{p_i}$ equal to the current location $\boldsymbol{x_i}$ in D-dimensional space.

5. Identify the particle in the neighbourhood with the best success so far, and assign its index to the global variable $p_g$.

6. Change the velocity and position of the particle according to the following equation (see notes below)

$$
\begin{cases}
\boldsymbol{v_i} \leftarrow \chi \left( \boldsymbol{v_i} + \boldsymbol{U}\left(0, \phi_1\right) \otimes \left(\boldsymbol{p_i} - \boldsymbol{x_i}\right) + \boldsymbol{U}\left(0, \phi_2\right) \otimes \left(\boldsymbol{p_g} - \boldsymbol{x_i}\right) \right), \\
\boldsymbol{x_i} \leftarrow \boldsymbol{x_i} + \boldsymbol{v_i}
\end{cases}
\tag{1}
$$

where

– $\chi$ is "constriction coefficients" to control the convergence of the particle

$$
\chi = \frac{2}{\phi - 2 + \sqrt{\phi^2 - 4\phi}}
\tag{2}
$$





where $\phi = \phi_1 + \phi_2 > 4$. $\phi$ is commonly set to 4.1, and $\phi_1 = \phi_2$. $\phi_1$ and $\phi_2$ are often referred to as the acceleration coefficients, which determine the magnitude of the random forces in the direction of the personal best ($pbest$) and the global best ($g$).

– $\boldsymbol{U}(0, \phi_i)$ is a vector of random numbers uniformly distributed in $[0, \phi_i]$, which is randomly generated at each iteration and for each particle.

    – $\otimes$ is a component-wise multiplication.

    – each component of $\boldsymbol{v_i}$ is kept within the range $[-V_{max}, +V_{max}]$ so that particles do not go out of search spaces. The optimal value of $V_{max}$ is problem-specific, but no reasonable rule of thumb is known. For this study, $V_{max}$ is

half of the maximum of the search space, or 0.5.

7. If the $v(x+1)$ potentially places $x(t+1)$ outside its defined search space, the out-of-bounds particle must be carefully repositioned. Imagine a ball (a particle) moving between 2 walls (search space) with velocity $v$. When the ball hits one of the walls, it bounces back to a position between the 2 walls. To represent this, choose the "damping" value, which controls the energy loss of the bouncing ball, to be something like $0 < \beta = 0.8 < 1$ (quite close to 1). If the particle

crosses the lower bound, then reposition the particle as $x(t+1) = r \cdot \beta \cdot x(t)$, where $r$ is a random number uniformly distributed between 0 and 1. So the particle has been randomly repositioned somewhere between its original position and the lower bound (but the damping ensures that it does not get too close to its original position). If the particle crosses the upper bound, then the same bouncing ball analogy applies, but the repositioning is $x(t+1) = 1 + r \cdot \beta \cdot (x(t) - 1)$. (Note that, this assumes that the parameter search space has been rescaled to the (0, 1) interval). The velocity of an

out-of-bounds particle should also be reset. The reset velocity vector should point away from the boundary, towards the original position $x(t)$. Sticking with the bouncing ball analogy: the velocity decreases with the distance bounced from the ground. So if the new position $x(t+1)$ is far from the boundary, the velocity will be small. Thus, velocities can be reset as: $v(t+1) = (r \cdot \beta - 1) \cdot v(t)$, which works for particles that cross either the upper or lower boundary.

8. When a criterion is met (usually a sufficiently good fitness or a maximum number of iterations), exit loop.

9. **end**

**The optimisation fitness function:** in this study, we use the Willmott skill score (WSS) (Willmott, 1981) mean absolute error ($WSS\_MAE$, equation 3) (Willmott et al., 2012) to evaluate the goodness of fit of the model to the data in the PSO. $WSS\_MAE$ ranges from 0 to 1, with values close to 1 indicating close agreement between model predictions and observations.

$$WSS\_MAE = 1 - \frac{\frac{1}{N}\sum_{i=1}^{i=N}|m_i - o_i|}{\frac{1}{N}\sum_{i=1}^{i=N}(|m_i - \bar{o}| + |o_i - \bar{o}|)} = 1 - \frac{MAE}{\frac{1}{N}\sum_{i=1}^{i=N}(|m_i - \bar{o}| + |o_i - \bar{o}|)} \tag{3}$$

where $m$ is model output, $o$ is observation, $N$ is number of pair of model–observation, and $\bar{o}$ is observations averaged.



**PSO with FABM:** the implementation of PSO with FABM is shown in the figure below

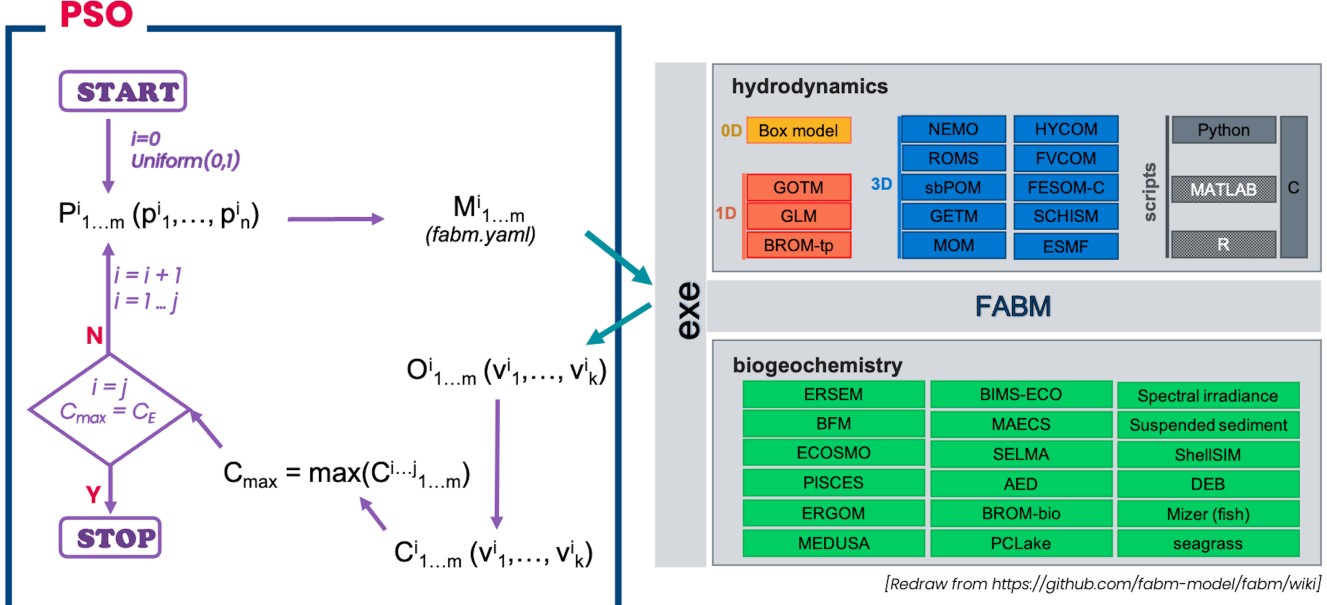

*[Redraw from https://github.com/fabm-model/fabm/wiki]*

**Figure 2.** Implementation of the Particle Swarm Optimiser (PSO) with FABM. First, the PSO toolbox initialises a set of model evaluations $(M^i_{1...m})$ by randomly assigning parameters $(P^i_{1...m})$ for each model evaluation in the set (iteration $i = 0$). The parameter values are restricted to the range [0, 1]. These parameters are then rescaled to their true ranges before being written to the model's `fabm.yaml` file. The toolbox then calls the executable file of the corresponding model or coupled model from FABM (for this study, GOTM-ECOSMO was used for demonstration) to run the model setup $M$. Upon completion of the model runs, the toolbox extracts the model outputs of the state variables corresponding to the observations $(O^i_{1...m}(v^i_1,...,v^i_k))$. It then calls the cost function to compute the skill score for each state variable $(C^i_{1...m}(v^i_1,...,v^i_k))$, identifying the maximum skill score $(C_{max})$ among all model evaluations in the set. The $C_{max}$ and its corresponding parameters represent the model that best fits the observations for that iteration. The $C_{max}$ and its corresponding parameters are then used as a reference point to calculate parameters for the set of model scores $(M^i_{1...m})$ in the next iteration. This process is repeated until the expected skill score $(C_E)$ or the maximum number of iterations is reached.

A quick guide to the PSO Toolbox: The use of the toolbox requires the input of several parameters. For example, it is necessary to specify the parameters to be calibrated with their respective ranges, the directory path to the executable file of a biogeochemical (BGC) model compiled in the Framework for Aquatic Biogeochemical Models (FABM), validated state variables and corresponding observations for the calculation of model skill scores. In addition, users must specify the number of iterations to run the PSO, the number of model evaluations per iteration, and the intervals and frequency for randomly resetting parameters rather than inferring them from parameters and skill scores from the previous iteration. It is essential that the cost function is adapted to specific requirements. Full details, including access to the toolbox code, manual and examples, can be found in the PSO Toolbox repository, available here.





## 2.2 The coupled 1D GOTM-ECOSMO E2E model

GOTM (General Ocean Turbulence Model, www.gotm.net) is an open-source community model for hydrodynamics and turbulent mixing processes in coasts, oceans and lakes (Umlauf et al., 2016). It is a one-dimensional water column model that can be run in stand-alone mode or coupled to a 3D circulation model. The core of the model computes solutions to the one-dimensional

versions of the transport equations of momentum, salt and heat.

ECOSMO E2E (Daewel et al. (2019)) is an extended ecosystem model that includes functional group formulations for fish and macrobenthos in addition to the state variables related to nutrients, phytoplankton, zooplankton and fish. The higher trophic levels in the original model were formulated for one functional group of fish, but the model code also includes a version with a two-group version previously used in an application for the North Atlantic (pelagic fish and demersal fish) and macrobenthos

(benthic suspension-/filter-feeders and benthic deposit-feeders). The biological processes of the model have been described in detail by (Daewel et al. (2019)) and its earlier versions (Daewel and Schrum (2013); Schrum et al. (2006)). In this version, the macrobenthos was adjusted to reflect the large amount of filter-feeding macrobenthos inhabiting the Sylt (Asmus and Asmus, 1991; Asmus, 2011). The model parameters subjected to calibration are provided in the Appendix (table A1 to A3).

The GOTM and ECOSMO E2E models are configured and coupled in FABM via yaml files (gotm.yaml for GOTM and

fabm.yaml for ECOSMO/ECOSMO E2E). The goodness of fit of the model is then accessed by the Willmott skill score (WSS) mean absolute error ($WSS\_MAE$, equation 3) (Willmott et al., 2012).

## 2.3 Model configuration and setup

**Study site:**The Sylt Road (55.03N, 8.46E) located in the Sylt-Rømø Bight (SRB), which is situated east of the islands of Sylt and Rømø ,is one of the large tidal basins of the Wadden Sea. The SRB is drained by three tidal inlets, the Rømø Dyb,

the Høyer Dyb and the Lister Tief, all three of which meet in the Lister Ley basin, which is connected to the North Sea by a narrow opening of 2.6 km between the islands Asmus (2011). Two rivers, Vidå and Bredeå, flow into the bay and drain a catchment area of about 1554km$^2$ (1081 km$^2$ and 473 km$^2$, respectively) Asmus (2011). Ecological research on fish and shellfish has been carried out in the SRB since early in 1930s, and in the 1990s extended ecosystem analyses were carried out to investigate material and energy flows in the SRB intertidal zone. While this research has added to our knowledge of Wadden

Sea ecosystems, an overall and common view of the interlinked dynamics of material flow and the organisms has not yet been achieved Asmus (2011). Two-dimensional hydrodynamic and numerical models described in the 1990s remained largely limited to abiotic processes such as currents and material transport (Stanev et al., 2003; Kohlmeier and Ebenhöh, 2009). In this study, we have proposed a model that describes the interlinked dynamics of organisms (or describes the dynamics of ecosystem processes), which will provide the necessary information and understanding to explore the hypotheses previously mentioned

in the Introduction and to provide input to other models such as Ecopath-Ecosim or ENA model (Horn et al., 2021a, b) to investigate the functioning of the ecosystem.

**GOTM forcings:** we use MERRA-2 (Modern-Era Retrospective analysis for Research and Applications, Version 2) meteorological data around the Sylt Road site. The data were downloaded from the NASA Goddard Earth Sciences (GES) Data



and Information Services Center (DISC). The meteorological data include temperature at 2 and 10 m; wind components at 2,
10 and 50 m; surface pressure and total precipitable water, surface albedo, cloud fraction, cloud optical thickness, incoming
surface shortwave flux (i.e. solar radiation), surface net downward shortwave flux and upward longwave flux at toa (top of
atmosphere), total precipitation, bias-corrected total precipitation, surface air temperature, surface specific humidity, surface
wind speed and turbulent evaporation. The data were then processed in the GOTM format.

**ECOSMO and ECOSMO E2E forcings and validation:** we use data monitored at Sylt Road (55.03N, 8.46E) for forcing,
boundary conditions and model validation. Specifically, we downloaded available data from www.pangaea.de from 2000 to
2008 (8 years of data). The weekly sampling dataset includes surface temperature (ºC), salinity (psu), nutrients (nitrate ($NO_3$,
$\mu$mol/l), phosphate ($PO_4$, $\mu$mol/l), silicate ($SiO_4$, $\mu$mol/l), turbidity (SPM), chlorophyll-a ($\mu$gChla/l), zooplankton (cell/l).
Details of data sampling methods and quality control are given in (Rick et al., 2023). Data were processed to the model unit
($mgC/m^3$ and $mgChla/m^3$) and written in the format required by GOTM and FABM. For zooplankton, we use the biomass per
cell given in (Martens, 1975) to convert the number of zooplankton per litter to biomass in carbon.

## 2.4 Model configuration and experimental parameterisation setups

The ECOSMO/ECOSMO E2E model setups were configured by modifying the corresponding fabm.yaml file. We performed
six experiments (detailed in Table 1) to demonstrate the PSO toolbox and to define a model that best describes the Sylt Road
ecosystem. These experiments ran from 2000 to 2004, with the first year being discarded as a spin-up period. The remaining
four years (2001 to 2004) were used for validation and computation within the PSO algorithm.

In these six experiments, we always calibrated phytoplankton-related parameters, including growth rates, mortality rates,
photosynthetic efficiency (aa) and light extinction (EXw), and gradually increased the complexity of the model and the num-
ber of parameters to be calibrated. Three experiments were carried out using ECOSMO, excluding the fish and macrobenthos
formulations. In the first experiment, we calibrated bottom-up processes using six parameters related to nutrients and light
(half saturation rates and water clarity). In the second experiment, we focused exclusively on top-down control processes and
calibrated 13 parameters related to zooplankton (grazing rates, half-saturation rates, mortality rates and food reference). The
third experiment combined both bottom-up and top-down processes, with a total of 19 parameters. The remaining three exper-
iments included macrobenthos and fish, and transitioned to the ECOSMO E2E model. In the fourth experiment, we calibrated
bottom-up and top-down processes as in the third experiment (calibration of 19 parameters), but included the macrobenthos
component without calibrating it. The fifth experiment followed the fourth, but included a macrobenthos calibration with 35
parameters. In the sixth experiment, we added a fish component and calibrated fish parameters, for a total of 65 parameters.

Details of the parameters tuned in each experiment, together with their corresponding ranges and default values, are given
in Tables A1 to A3 in the appendix. For each PSO configuration, the number of model evaluations per PSO iteration increased
with the number of calibrated parameters. In general, the number of model evaluations per iteration was at least twice the
number of calibrated parameters to ensure algorithm efficiency (Poli et al., 2007). Thus, the number of model evaluations per
iteration for each experiment was as follows Experiment (1): 30, (2): 45, (3) and (4): 65, (5): 90, and (6): 150. It should be noted



that increasing the number of calibrated parameters (or the number of model evaluations required per iteration) correspondingly increases the duration of the PSO run.

| Model type | Exp. No. | Experiment name | Parameters to tun |
|---|---|---|---|
| ECOSMO | 1 | bottom-up (**bt**) | 1 - 12 (12 parameters) as in table A1 |
| | 2 | top-down (**tdz**) | 1-4, 6-7, 13-25 (19 parameters) as in table A1 |
| | 3 | bottom-up and top-down (**bttdz**) | 1 - 25 (25 parameters) as in table A1 |
| ECOSMO E2E | 4 | bottom-up and top-down (**bttdz_e2e_nfi**) | 1 - 25 (25 parameters) as in table A1 |
| | 5 | bottom-up, top-down, and filter-feeder (**bttdzff_e2e_nfi**) | 1 - 41 (41 parameters) as in table A1 and A2 |
| | 6 | tuning all (**e2e_tuned_all**) | 1 - 65 (65 parameters) as in table A1, A2, and A3 |

**Table 1.** Description of parameterisation experiments. Six optimisation experiments were conducted in this study to identify the model that best represents the ecosystem around Sylt Road. Parameters related to phytoplankton growth (parameters 1-4 and 6-7 as listed in table A1) were calibrated in all six experiments. Experiments 1-3 were run with the ECOSMO model (an NPZD-type model), while experiments 4-6 were run with the ECOSMO E2E model, which includes macrobenthos and fish. Experiment 1, named *bt*, investigated bottom-up control by calibrating parameters related to nutrients and light. Experiment 2, named *tdz*, investigated top-down control by calibrating parameters related to zooplankton. Experiment 3, named *bttdz*, investigated both bottom-up and top-down controls. Experiment 4, named *bttdz_e2e_nfi*, extended experiment 3 by including macrobenthos in the model. Experiment 5, named *bttdzff_e2e_nfi*, was similar to experiment 4, but with calibration of macrobenthos-related parameters. Finally, experiment 6, named *e2e_tuned_all*, included the calibration of all model parameters.





## 3 Results and Discussions

### 3.1 Demonstration of the PSO toolbox

**Parameters identification and model skill score improvement**

In optimisation methods, it is generally expected that the optimiser runs until all parameters have converged. However, such runs are computationally demanding, especially when optimising a large number of parameters (more than 10). Numerous test simulations carried out during this study have shown that reliable results can be obtained from PSO well before full parameter convergence. In Appendices A1 and A2, we show that the skill scores of the model and the validated state variables, as well as the converged parameter values, stabilise around the 300th iteration. Therefore, we chose the 300th iteration as the stopping condition for the PSO in the six experiments shown in Table 1.

The results of the 6th experiment (e2e_tuned_all), shown in Figure 3(a), illustrate the effectiveness of the PSO toolbox. The skill scores of the validated variables—chlorophyll-a (Chla), small zooplankton (Zos), large zooplankton (Zol), nitrate ($NO_3$), phosphate ($PO_4$), and silicate ($SiO_4$) - as well as with the overall model skill score, calculated as the sum of the validated variables, show significant improvement throughout the PSO iterations. In particular, the overall model improvement (shown by the black line) occurred mainly within the first 50 iterations. From the 50th to the 100th iteration, the model skill score showed marginal increases, with the scores stabilising beyond the 100th iteration.

Parallel to the overall model improvement, the skill score improvement for the validated variables is mainly observed within the first 50 iterations. However, between the 80th and 100th iteration, while the overall model skill score shows a slight increase, the skill score of chlorophyll-a shows a significant increase. This increase is partially offset by a decrease for other variables. In particular, among the validated variables, the most significant improvements are seen in $PO_4$, SiO and large zooplankton, followed by Chl-a and small zooplankton. Conversely, $NO_3$ shows minimal improvement.





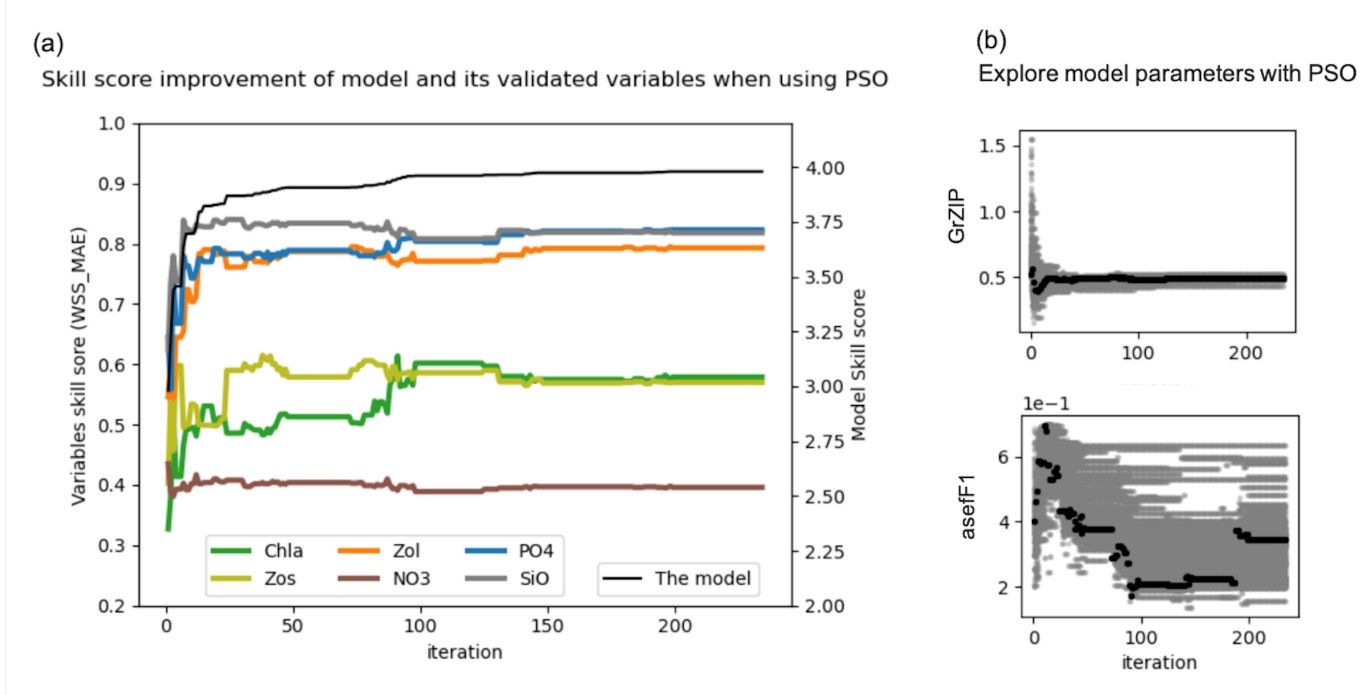

**Figure 3.** (a) Improvement of the skill score (WSS_MAE) of the coupled 1D GOTM-ECOSMO E2E model and its validated state variables over PSO iterations: The x-axis represents iterations, the left y-axis shows the skill scores of the validated state variables (Chla, small zooplankton - Zs, large zooplankton - Zl, nitrate - $NO_3$, phosphate - $PO_4$ and silicate - $SiO_4$) and the right y-axis shows the overall model skill score (black line), which is the sum of all validated state variables. (b) Exploration of model parameters through PSO iterations: the x-axis represents iterations and the y-axis represents parameter values. Grey dots indicate the parameter values explored, while black dots represent the best parameter values at each iteration. The figure illustrates parameter exploration for two parameters, *GrZlP* (grazing of large zooplankton on phytoplankton) and *asefF1* (assimilation efficiency of pelagic fish), as examples. The parameter exploration may converge quickly (e.g. *GrZlP*) or take longer to converge (e.g. *asefF1*).

In the present experiment (the 6th experiment), 65 parameters were tuned to refine the model. Examples of parameter
exploration are shown for the grazing rate of large zooplankton on phytoplankton (*GrZlP*) and the assimilation rate of pelagic fish (*asefF1*) (Figure 3 (b)). Details of parameter exploration for all parameters can be found in the Appendix (Figure A3). The parameters will eventually converge if the optimisation is run long enough, but reliable optimised parameters can often be obtained much earlier, before full convergence. Figure 3 (b) illustrates that the parameters can converge quickly (e.g. in the case of *GrZlP*) or take much longer to converge (e.g. in the case of *asefF1*). The speed of convergence is likely to depend
on the availability of relevant data used in the cost function. However, we cannot draw definitive conclusions about the speed of convergence. Our parameter exploration (Figure A3) showed that some parameters relevant to the available validation data converged rapidly (e.g. *muPl* (maximum growth rate of large phytoplankton), *GrZlP* (grazing rate of mesozooplankton on phytoplankton), *GrZsP* (grazing rate of microzooplankton on phytoplankton)), while others relevant to the same data converged




slowly (e.g. *mPl* (mortality rate of large phytoplankton), *rPO4* (half-saturation rate of phosphate), *rSi* (half-saturation rate

of silicate)). In addition, certain parameters without relevant validation data also showed rapid convergence (e.g. *GrF1MB2* (grazing rate of pelagic fish on deposit feeder), *rMB1* (half-saturation rate of filter feeder), *rMB2* (half-saturation rate of deposit feeder)). Further investigation is likely to be required to clarify the relationship between the speed of parameter convergence and the availability of validation data.

**Potential model parameter ranges**

As shown in Figure 3(a), the model skill score stabilises after the 100th iteration, although small fluctuations in the skill score of the validated variables persist over the next 50 iterations. This suggests that the parameter set from the 100th iteration provides a similar goodness of fit for the model. We have extracted the best parameter values from this iteration (black dots), as shown for *GrZlP* and *asefF1* in Figure 3(b), and plotted these values in Figure 4. It can be seen that about two thirds of the tuned parameters converge on certain values. The parameters with the largest ranges are predominantly associated with

fish or macrobenthic elements, which represent the highest trophic levels and ultimately the closure terms within the model. The use of closure terms to address model-data mismatch is a common practice in modelling. In addition, the lack of data to constrain the macrobenthic and fish parameters allows these parameters to vary freely during parameterisation, resulting in greater uncertainty. Overall, Figure 4 shows that the PSO toolbox can define specific values for some parameters and suggest possible ranges for others.

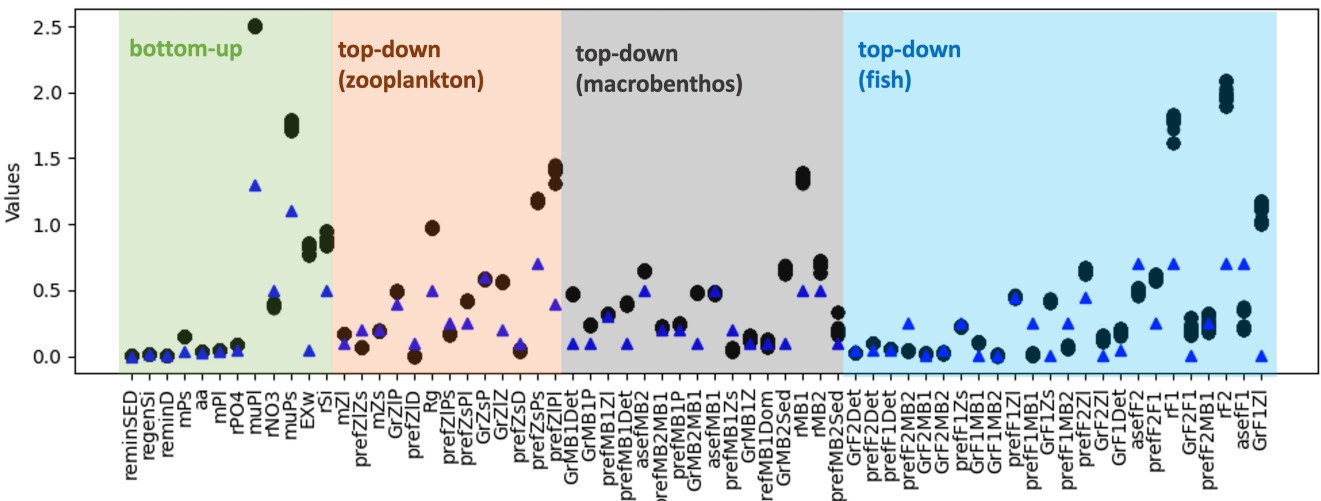

**Figure 4.** Potential parameter values and ranges suggested by the PSO toolbox are shown. The optimal parameter values from the 100th iteration (indicated by black dots), as shown in Figure 3(b) for *GrZlP* and *asefF1*, have been selected and plotted here. Blue triangle points are default parameter values of the model.



## PSO optimised parameter set for the ecosystem around Sylt Road

As the model skill score stabilised from the 100th iteration, we selected the parameter values from the 100th iteration as the optimal parameter set for the coupled 1D GOTM-ECOSMO E2E model to simulate the ecosystem around Sylt Road. We ran two models: one with the parameter set from the 100th iteration (named *model_pso*) and one with the default parameters (shown as blue triangle points in Figure 4) before applying PSO (named *model_ref*), for the period from 2000 to 2008. The year 2000 was excluded as a spin-up period. The period from 2001 to 2004 (shown in the shaded part of figure 5) was used in the PSO experiments (1). Consistent with the improvement in skill scores, the *model_pso* output for PO$_4$ and large zooplankton (mesozoo) (represented by the black line) over the period 2001 to 2004 closely matches the observational data (represented as grey dots). In addition, the model with the optimal parameter set from PSO captures the seasonal cycle of SiO better, although to a slightly lesser extent. The model effectively reproduces the spring phytoplankton bloom, but falls short in reproducing the lower magnitudes at other times of the year. In particular, the model successfully reproduces observations of small zooplankton (microzoo). Compared to the default model before the application of PSO (blue line), the model with the parameter set found by PSO shows a significantly better performance.







**Figure 5.** Model outputs (black line) using the parameter set found by PSO at iteration 100 and observations (solid grey dots) from 2001 to 2008, where 2001 to 2004 was the tuning period and 2005 to 2008 showed how well the parameter set worked in other periods.

As indicated by the $NO_3$ skill score, the model faces challenges in resolving $NO_3$ to match observations. While the model reproduces the seasonal cycle of $NO_3$, it fails to capture both the magnitude and the period of $NO_3$ removal. This discrepancy arises because the model setup does not account for $NO_3$ input from rivers flowing into the Sylt-Romo Bight. In addition, the one-dimensional (1D) model does not account for horizontal advection, which can transport material to and from the site. Therefore, the discrepancy in $NO_3$ is likely due to the limitations of the 1D model setup and cannot be resolved by parameterisation alone. Analysis of the model output for the period 2005 to 2008 (the unshaded part of Figure 5) shows that the optimal parameters identified for the period 2001 to 2004 continue to perform well over a longer period. This indicates that the parameter set is robust or that the simulated period (2005-2008) has a similar cycle to the previously parameterised period (2001-2004).




## 3.2 Multiple parameter sets can reproduce observations equally well despite differences in values

As shown in the previous section, a large number of parameter sets reproduce the observational data equally well. To explore
these parameter sets in more detail, we focus on model runs using parameters extracted from iterations 100 to 300 at intervals
280 of 5 (see Figure 5). Despite notable differences in parameter values, such as *asefF1*, *GrF1Zl*, *GrF2F1* and *rSi*, the skill score
and validated state variables of the model remain relatively unchanged (see Figure 3). The outputs generated by these param-
eter sets show almost identical characteristics, with minor variations observed in ammonia (NH$_4$) and flagellate biomass (in
chlorophyll *a*). Notably, significant parameter differences occur mainly in fish related parameters (specifically *asefF1*, *GrF1Zl*
and *GrF2F1*). The slight increase in *rSi*, the half-saturation rate of silicate, is not sufficient to significantly affect phytoplankton
285 biomass. Fish biomass is close to zero in all model outputs (not shown), due to the PSO setting its half saturation rate high,
resulting in minimal growth. While acknowledging the inaccuracies of this approach, the lack of data to inform fish-related pa-
rameters, compounded by the complexity of the closure terms as discussed above, contributes to the unresolved discrepancies
between model and observational data.

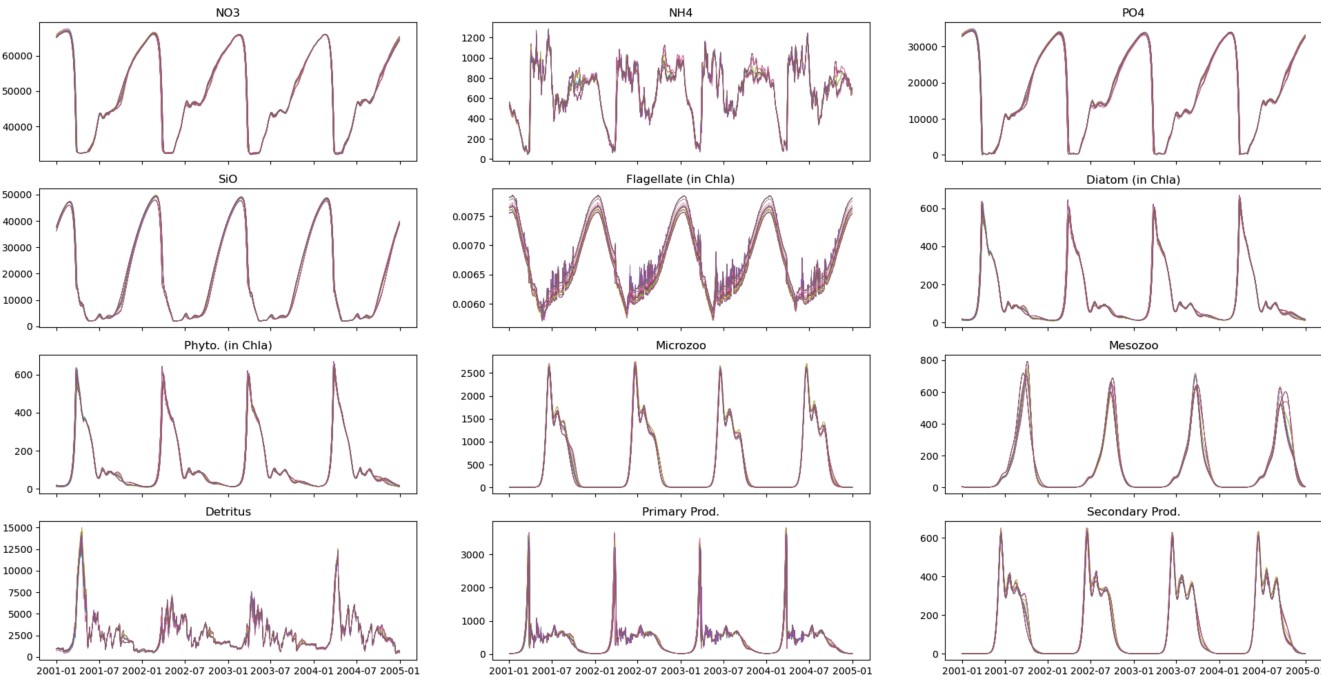

**Figure 6.** Multiple parameter sets can reproduce observations equally well despite differences in their values. The lines represent the output
of 40 model runs, each using a different parameter set, evaluated between the 100th and 300th iteration at 5 interval. Although these parameter
sets differ in their values (Figure A3), they produce similar results.

In this study, no differences were observed between the model runs despite variations in their parameter sets. In a similar
290 study using Particle Swarm Optimisation (PSO) to parameterise a 1D biogeochemical model of Puget Sound (Nguyen, 2021),
the author reported marked differences in model output between parameter sets that achieved equally good fits to observations.





The results of the Puget Sound study are reprinted here in an Appendix to highlight instances where PSO can elucidate different systems capable of reproducing identical observations. The Puget Sound study examined two parameter sets that differed in phytoplankton growth (growth rate, $I_0$, mortality), zooplankton (mortality), and detritus (sinking) (see Figure A4), resulting in notable disparities in carbon transfer between trophic levels (via grazing) and carbon export (via sinking) (see Figure A5). These disparities imply that systems with completely different dynamics can yield equivalent goodness of fit to observations.

Thus, optimised parameter sets can result in either identical (as demonstrated in this study) or different (as observed in the Puget Sound study) system dynamics. In either case, the optimisation provides an insight into the potential of the system and the potential buffer that the parameters can take to maintain the system stability.

## 3.3 Optimising model complexity with PSO: top-down control by macrobenthos in the marine ecosystem around Sylt Road.

Figure 7 shows the model outputs resulting from runs using the best parameter sets identified by the PSO for six experiments detailed in Table 1. The black line represents the sixth experiment, previously shown in Figure 5, which shows a good fit to the observations. In particular, the first experiment (or bottom-up control experiment, *bt*) shows the inability of the PSO to find bottom-up parameters capable of matching the model to the observations, indicating that tuning the bottom-up process alone proves insufficient to achieve model accuracy. Subsequent experiments progressively introduce modifications: the second experiment (*tdz*) involves tuning top-down (zooplankton) control parameters, leading to a partial improvement in model fit, but still falling short of good agreement with observations. The third experiment (*bttdz*), which tunes both bottom-up and top-down processes, shows only marginal improvement over the second experiment, highlighting the predominant role of top-down control in improving model fitness. The fourth experiment (*bttdz_e2e_nfi*), which incorporates macrobenthos (filter-feeders and sediment-feeders) without tuning, closely resembles the results of the sixth experiment, suggesting the critical influence of macrobenthos in closely reproducing observations. This finding is consistent with the presence of a substantial mussel bed in the Sylt-Romo Bight ecosystem, which is known to significantly influence phytoplankton and trophic dynamics of the Bight ecosystem (Asmus and Asmus, 1991; Baird et al., 2007) . Subsequent experiments (fifth and sixth), involving additional tuning of macrobenthos-related parameters and the inclusion of fish, respectively, yield only marginal improvements over the fourth experiment.

Overall, our analysis of these six experiments shows that the ECOSMO model (NPZD style model) combined with macrobenthos provides an optimal representation of the Sylt Road ecosystem. This finding underlines that increasing model complexity, such as the inclusion of fish, does not necessarily improve performance in this context. Previous studies (Huse and Fiksen, 2010; Fulton et al., 2003; Friedrichs et al., 2007; Fulton, 2010) have also suggested that adding complexity to a model is not always beneficial or necessary. Kuhn and Fennel (2019) has also emphasised the importance of identifying the appropriate level of complexity in ecosystem models to achieve meaningful representations of biogeochemical cycles across spatial and temporal scales.





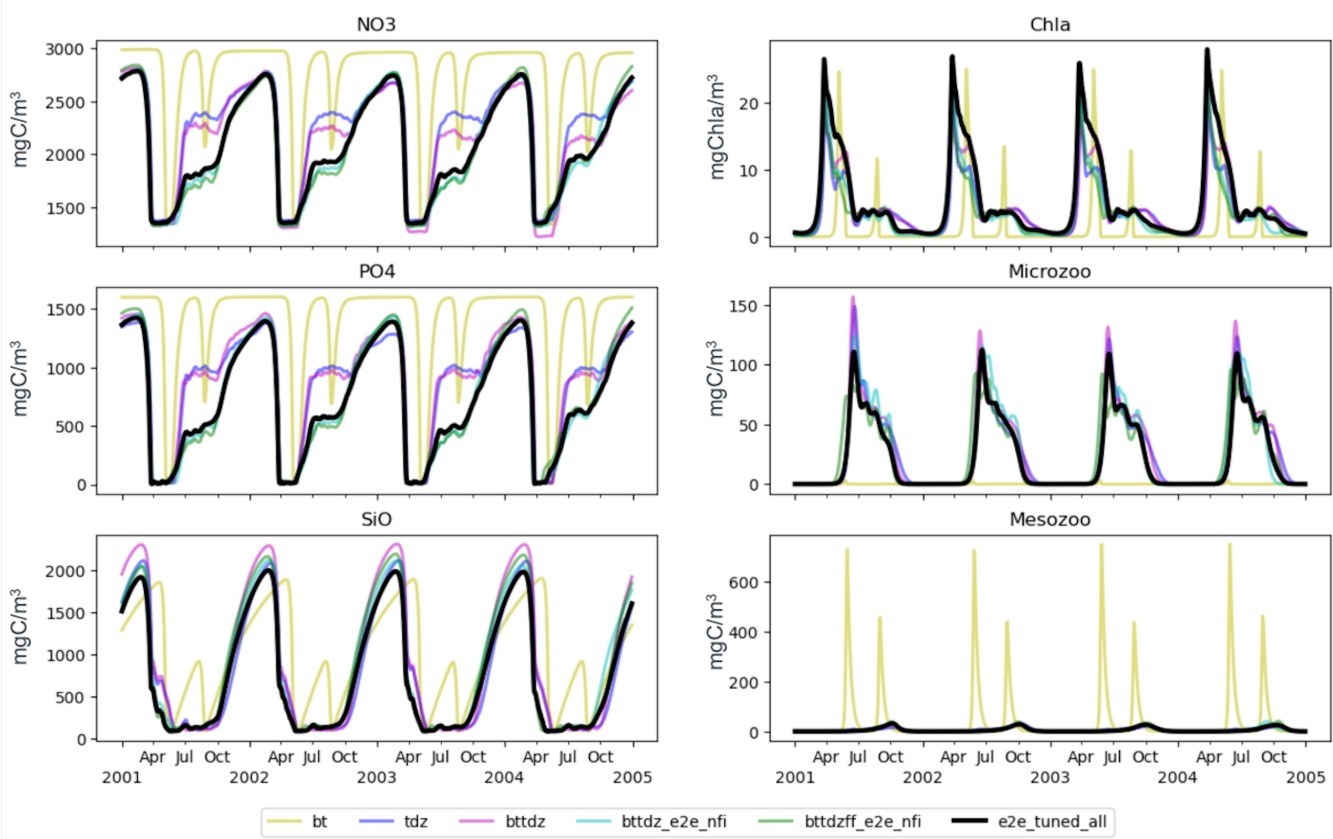

**Figure 7.** Model outputs of six PSO experiments described in table 1. The black line represents the 6th experiment, tune all parameters (e2e_tuned_all), which was previously shown in figure 5. The model outputs of the parameters from the 6th experiment showed a good fit to the observations.

## 4 Conclusions

325 The Particle Swarm Optimizer (PSO) toolbox has demonstrated its effectiveness in identifying parameter values that generally reproduce the Sylt observations, with the exception of the nitrate magnitude. This discrepancy is probably due to the limitations of the 1D model in resolving the hydrodynamic complexity in the Sylt-Romo Bight (e.g. tidal flats) and the lack of river inputs in the model setup. The toolbox, written in Fortran 90, has proven to communicate effectively with models or coupled models within FABM by simply invoking the model executable (created after compilation) in FABM.

330 While some adjustments to the toolbox code, such as the cost functions (or model skill score) and other factors, may be required due to variations in FABM model outputs and simulation periods, the underlying algorithm remains adaptable. Detailed instructions on what and where to make changes to the toolbox to adapt it to new model simulations are can be found in the toolbox repository at https://codebase.helmholtz.cloud/hoa.nguyen/parameterisation.





In addition to parameter identification, we have shown that the parameterisation is able to find alternative Sylt ecosystems that fit the observational data equally well, providing different perspectives on the ecosystem under study. Furthermore, the toolbox is able to define an optimal model for the ecosystem around the Sylt Road.

Despite its robustness, the effectiveness of the toolbox depends on the availability of data to identify model parameters and the computational resources required to run the algorithm. The toolbox is currently suitable for fast running models, with execution times ranging from seconds to several minutes. However, it is not well suited to large models with higher computational requirements, such as those that take hours to run. For these models, advanced methods may be required, highlighting the continuing need for advances in optimisation techniques.

In summary, the Toolbox has the following advantages:

1. Improved goodness of fit of marine ecosystem models: The application of the Particle Swarm Optimiser (PSO) to biogeochemical modelling leads to significant improvements in model accuracy and overcomes the challenges associated with traditional parameterisation techniques (try-and-error).

2. Reusability and accessibility: The implementation of a PSO compatible with the Framework for Aquatic Biogeochemical Models (FABM) opens up opportunities for further optimisation applications to models in the FABM. This will encourage the practice of model parameterisation to improve model accuracy.

3. Insights into model complexity and versatility: The study of model complexity suggests the optimal representation of an ecosystem. The identification of different parameter sets with equally good fits to observations demonstrates the robustness and versatility of the toolbox.

and limitations:

1. The parameter set is only valid for the system under consideration. It may be applicable to nearby systems with similar environmental and hydrodynamic conditions. However, for systems with different conditions, re-parameterisation is necessary. A colleague of ours attempted to apply the Sylt parameter set to an ecosystem in the central North Sea, but failed to reproduce the ecosystem accurately. This failure is attributed to the differences in the light climate between the central and shallow coastal North Sea.

2. Data constraints and model limitations: A major challenge in ecosystem modelling is the availability and quality of data required to inform and constrain the model. This limitation is not unique to the method presented, but is a common obstacle in all modelling efforts. The accuracy of the parameterisation toolbox is inevitably linked to the completeness and reliability of the data available.

3. The challenge of computational efficiency arises with the number of tuned parameters. While the Particle Swarm Optimiser (PSO) toolbox demonstrates efficiency in parameterising models, it is important to recognise its limitations in terms of computational time. The runtime of the toolbox depends on the number of parameters to be tuned, as more model evaluations are required. For example, the 1st experiment (12 parameters) took 17 hours to run, while the 6th





experiment (65 parameters) took 5 days. Although it is suitable for models with a moderate number of parameters, its applicability decreases when dealing with an extensive parameter set. This limitation requires a pragmatic consideration of model complexity and a judicious selection of parameters for optimisation to ensure computational feasibility.

4. Speed dependence on model runtime: a notable limitation of the PSO toolbox is its sensitivity to model runtime. While the method performs exceptionally well with fast running models (i.e., seconds to few minutes), it faces significant challenges with slower running models (i.e., hours or days). Although the theoretical feasibility of the method remains intact, its practicality is significantly reduced due to longer execution times. This issue highlights the need for methodological adaptations or alternative approaches when dealing with models characterised by extended run time.

5. This study examines the relationship between observations, constrained/unconstrained parameters, the number of parameters selected for optimisation and the level of model complexity is examined in this study. In Section 3.1, we were unable to explain why some parameters converged earlier than others in terms of observation availability. We suggest that further investigation of the relationship between observation availability and parameter constraints, as well as the effect of the number of parameters chosen for optimisation and model complexity, is needed. This would provide a comprehensive guide to model optimization.

Future work: We plan to undertake the following tasks to address the current limitations of the toolbox.

1. Further investigation of the relationship between observational constraints and parameter values, as well as the number of parameters selected for optimisation and model complexity, is needed to provide comprehensive guidance on model optimisation.

2. Application of optimisation to 3D biogeochemical models: As mentioned above, the current toolbox is suitable for 1D (or fast running) models, but is not yet practical for 3D models. Although the toolbox theoretically works for 3D models, the runtime is currently impractical. We plan to improve our existing toolbox to increase its speed and are also open to incorporating new methods suitable for 3D optimisation into the toolbox.

*Code availability.* the toolbox and example are available at DOI: https://doi.org/10.5281/zenodo.13904053. Learn more about PSO toolbox and view the latest version at https://codebase.helmholtz.cloud/hoa.nguyen/parameterisation





390 **Appendix A**

**A1**

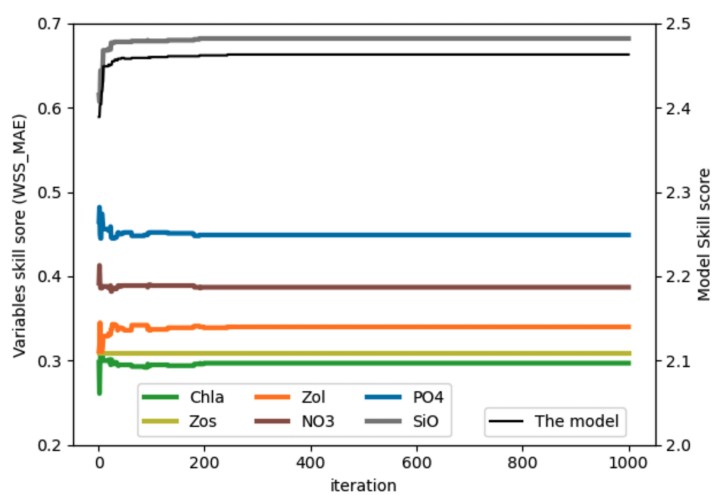

**Figure A1.** Model and its variable skill scores of the 1st experiment with PSO. The experiment illustrated that the skill scores highly improve in the first 50 iteration, after that they are only slightly improved, and around the 200th iteration, the skill scores are stabilized even though parameters are still fluctuated much longer.





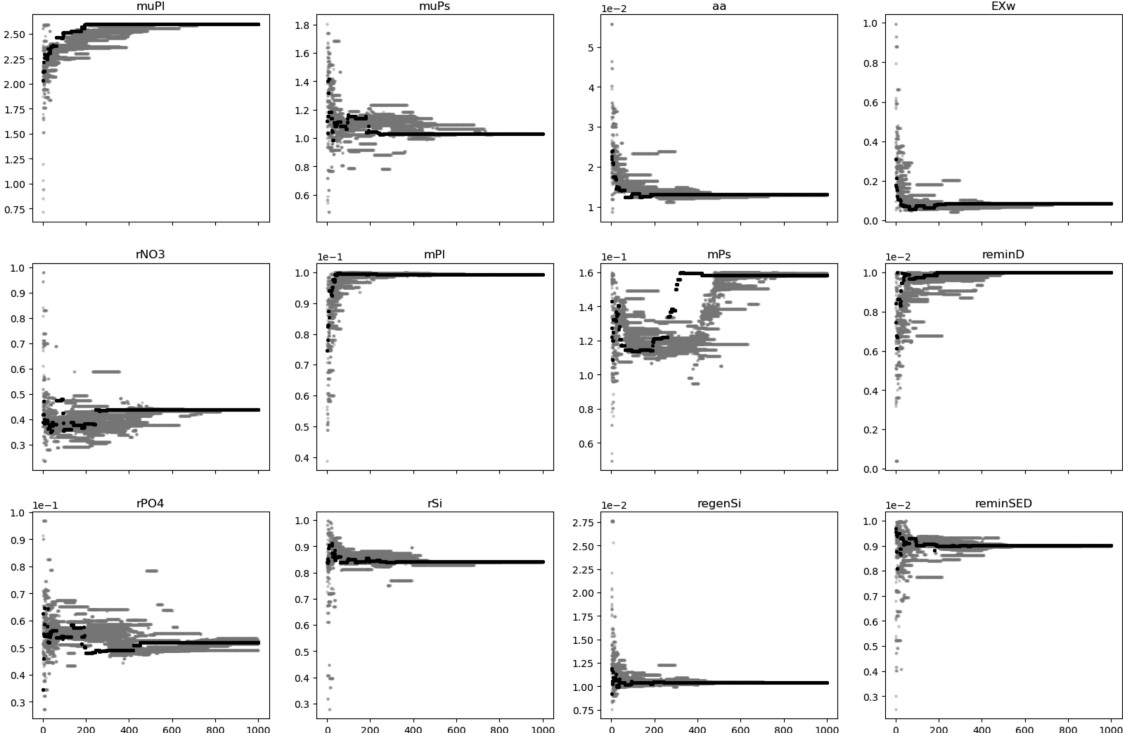

**Figure A2.** PSO parameters exploration of the 1st experiment. x-axis is iteration, and y-axis is parameter value. In a given iteration, gray dots are parameters values tried and black dot is the parameters values that give a model best fit to observation at the iteration. This experiment illustrated that a reliable optimised parameter set (as an output of the PSO) can be achieved much earlier (i.e., around the 300th iteration) than at the converged point (e.g., the 1000th iteration).





**Figure A3.** PSO parameters exploration of the 6th experiment. x-axis is iteration, and y-axis is parameter value. In a given iteration, gray dots are parameters values tried and black dot is the parameters values that give a model best fit to observation at the iteration.



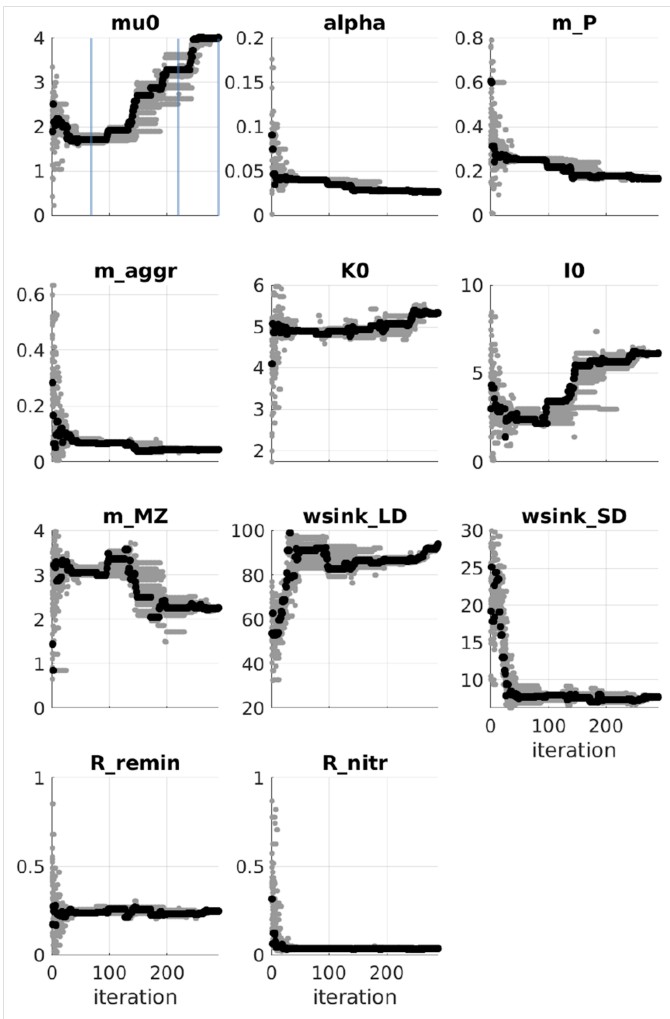

**Figure A4.** PSO for Puget Sound ecosystem model (Nguyen (2021)). Parameters exploration: In a given iteration, gray dots are parameters values tried and black dot is the parameters values that give a model best fit to observation at the iteration. Blue lines mark the iteration number 70, 220, and 300. (The figure is used with permission)



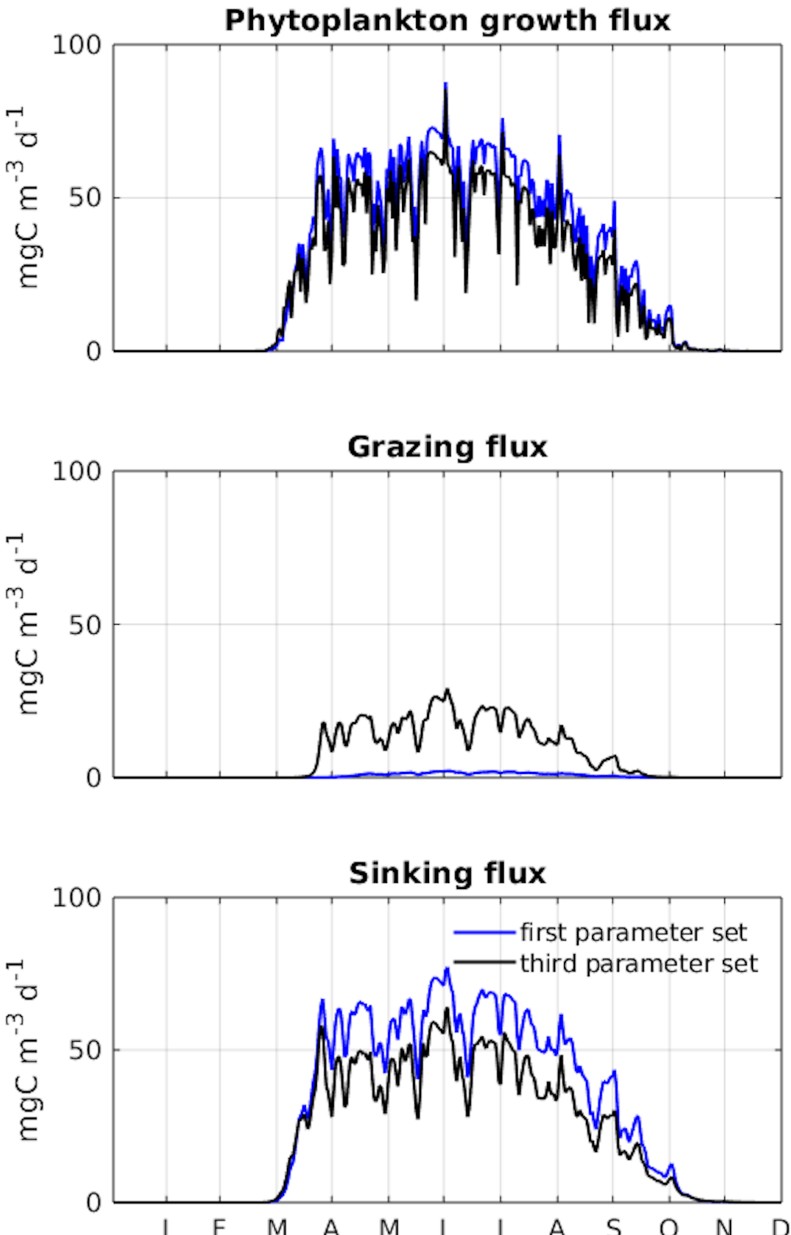

**Figure A5.** PSO for Puget Sound ecosystem model (Nguyen (2021)). Differences in dynamics between the two parameter sets which produce similar model goodness-of-fit. The first and third parameter sets are taken at the 70th and 220th iteration as marked in figure A4. The figure shows clear difference in carbon transfer between trophic levels (via grazing) and carbon export (via sinking) between the two parameter sets. (The figure is used with permission)



|  | Short | Long name | Unit | Reference value | Range |
|---|---|---|---|---|---|
| 1 | $muPl$ | max growth rate for large Phytoplankton | $1/day$ | 1.3 | [0.6, 2.6] |
| 2 | $muPs$ | max growth rate for small Phytoplankton | $1/day$ | 1.1 | [0.5, 2.2] |
| 3 | $aa$ | Photosynthesis efficiency | $m^2/W$ | 0.03 | [0.01, 0.06] |
| 4 | $EXw$ | Light extinction | $1/m$ | 0.05 | [0.02, 1.0] |
| 5 | $rNO3$ | NO3 half saturation | $mmolN/m^3$ | 0.5 | [0.25, 1.0] |
| 6 | $mPl$ | Pl mortality rate | $1/day$ | 0.04 | [0.02, 0.1] |
| 7 | $mPs$ | Ps mortality rate | $1/day$ | 0.08 | [0.04, 0.16] |
| 8 | $reminD$ | Detritus remin. rate | $1/day$ | 0.003 | [0.001, 0.006] |
| 9 | $rPO4$ | PO4 half saturation | $mmolP/m^3$ | 0.05 | [0.025, 0.1] |
| 10 | $rSi$ | SiO2 half saturation | $mmolSi/m^3$ | 0.5 | [0.25, 1.0] |
| 11 | $regenSi$ | Si regeneration rate | $1/day$ | 0.015 | [0.007, 0.03] |
| 12 | $reminSED$ | sediment remineralization rate | $1/day$ | 0.001 | [0.0005, 0.01] |
| 13 | $GrZlP$ | Grazing rate Zl on Phyto | $1/day$ | 0.8 | [0.4, 1.6] |
| 14 | $GrZsP$ | Grazing rate Zs on Phyto | $1/day$ | 1.0 | [0.5, 2.0] |
| 15 | $GrZlZ$ | Grazing rate Zl on Zs | $1/day$ | 0.5 | [0.25, 1.0] |
| 16 | $Rg$ | Zs, Zl half saturation | $mmolN/m^3$ | 0.5 | [0.25, 1.0] |
| 17 | $mZl$ | Zl mortality rate | $1/day$ | 0.1 | [0.05, 0.2] |
| 18 | $mZs$ | Zs mortality rate | $1/day$ | 0.2 | [0.1, 0.4] |
| 19 | $prefZsPs$ | Grazing preference Zs on Ps | - | 0.7 | [0.3, 1.4] |
| 20 | $prefZsPl$ | Grazing preference Zs on Pl | - | 0.25 | [0.12, 0.5] |
| 21 | $prefZsD$ | Grazing preference Zs on Det | - | 0.0 | [0.001, 0.4] |
| 22 | $prefZlPs$ | Grazing preference Zl on Ps | - | 0.1 | [0.05, 0.3] |
| 23 | $prefZlPl$ | Grazing preference Zl on Pl | - | 0.85 | [0.4, 1.7] |
| 24 | $prefZlZs$ | Grazing preference Zl on Zs | - | 0.15 | [0.07, 0.4] |
| 25 | $prefZlD$ | Grazing preference Zl on Det | - | 0.0 | [0.002, 0.4] |

**Table A1.** ECOSMO and ECOSMO E2E model parameters (part 1)





|    | Short | Long name | Unit | Reference value | Range |
|----|-------|-----------|------|-----------------|-------|
| 26 | $rMB1$ | MB filter feeder mortality rate | $1/day$ | 0.5 | [0.25, 1.5] |
| 27 | $asefMB1$ | MB filter feeder assimilation efficiency | - | 0.5 | [0.3, 0.8] |
| 28 | $GrMB1Z$ | Grazing rate MB1 on Zooplankton | $1/day$ | 0.1 | [0.01, 0.5] |
| 29 | $GrMB1P$ | Grazing rate MB1 on Phytoplankton | $1/day$ | 0.1 | [0.01, 0.5] |
| 30 | $GrMB1Det$ | Grazing rate MB1 on Detritus | $1/day$ | 0.1 | [0.01, 0.5] |
| 31 | $prefMB1P$ | Grazing preference MB1 on P | - | 0.2 | [0.01, 0.5] |
| 32 | $prefMB1Zs$ | Grazing preference MB1 on Zs | - | 0.2 | [0.01, 0.5] |
| 33 | $prefMB1Zl$ | Grazing preference MB1 on Zl | - | 0.3 | [0.01, 0.5] |
| 34 | $prefMB1Det$ | Grazing preference MB1 on Det | - | 0.1 | [0.01, 0.5] |
| 35 | $prefMB1Dom$ | Grazing preference MB1 on Dom | - | 0.1 | [0.01, 0.5] |
| 36 | $rMB2$ | MB deposit feeder grazing half saturation | $mmolN/m^3$ | 0.5 | [0.25, 2.0] |
| 37 | $asefMB2$ | MB deposit feeder assimilation efficiency | - | 0.35 | [0.2, 0.7] |
| 38 | $GrMB2Sed$ | Grazingrate MB2 on Sediment | $1/day$ | 0.1 | [0.05, 1.5] |
| 39 | $GrMB2MB1$ | Grazingrate MB2 on MB1 | $1/day$ | 0.1 | [0.01, 0.5] |
| 40 | $prefMB2Sed$ | Grazing preference MB2 on Sed | - | 0.1 | [0.05, 0.8] |
| 41 | $prefMB2MB1$ | Grazing preference MB2 on MB1 | - | 0.2 | [0.001, 0.5] |
| 42 | $rF1$ | fish1 grazing half saturation | $mmolN/m^3$ | 0.7 | [0.35, 2.0] |
| 43 | $asefF1$ | fish1 assimilation efficiency | - | 0.7 | [0.2, 1.0] |
| 44 | $GrF1Zl$ | Grazing rate fish1 on Mesozoo | $1/day$ | 0.01 | [0.005, 1.5] |
| 45 | $GrF1Zs$ | Grazing rate fish1 on Microzoo | $1/day$ | 0.01 | [0.005, 0.7] |

**Table A2.** ECOSMO and ECOSMO E2E model parameters (part 2)



|    | Short | Long name | Unit | Reference value | Range |
|----|-------|-----------|------|-----------------|-------|
| 46 | $GrF1Det$ | Grazing rate fish1 on Det | $1/day$ | 0.05 | [0.001, 0.5] |
| 47 | $GrF1MB1$ | Grazing rate fish1 on MB1 | $1/day$ | 0.01 | [0.001, 0.5] |
| 48 | $GrF1MB2$ | Grazing rate fish1 on MB2 | $1/day$ | 0.01 | [0.001, 0.5] |
| 49 | $prefF1Zl$ | Grazing preference fish1 on Zl | - | 0.45 | [0.3, 0.7] |
| 50 | $prefF1Zs$ | Grazing preference fish1 on Zs | - | 0.25 | [0.001, 0.3] |
| 51 | $prefF1Det$ | Grazing preference fish1 on Det | - | 0.05 | [0.001, 0.1] |
| 52 | $prefF1MB1$ | Grazing preference fish1 on MB1 | - | 0.25 | [0.001, 0.5] |
| 53 | $prefF1MB2$ | Grazing preference fish1 on MB2 | - | 0.25 | [0.001, 0.5] |
| 54 | $rF2$ | fish2 grazing half saturation | $mmolN/m^3$ | 0.7 | [0.35, 2.3] |
| 55 | $asefF2$ | fish2 assimilation efficiency | - | 0.7 | [0.1, 1.0] |
| 56 | $GrF2Zl$ | Grazing rate fish2 on Mesozoo | $1/day$ | 0.01 | [0.005, 1.5] |
| 57 | $GrF2F1$ | Grazing rate fish2 on Microzoo | $1/day$ | 0.01 | [0.005, 1.5] |
| 58 | $GrF2Det$ | Grazing rate fish2 on Det | $1/day$ | 0.05 | [0.001, 0.05] |
| 59 | $GrF2MB1$ | Grazing rate fish2 on MB1 | $1/day$ | 0.01 | [0.001, 0.1] |
| 60 | $GrF2MB2$ | Grazing rate fish2 on MB2 | $1/day$ | 0.01 | [0.001, 0.1] |
| 61 | $prefF2Zl$ | Grazing preference fish2 on Zl | - | 0.45 | [0.1, 0.7] |
| 62 | $prefF2F1$ | Grazing preference fish2 on F1 | - | 0.25 | [0.1, 0.7] |
| 63 | $prefF2Det$ | Grazing preference fish2 on Det | - | 0.05 | [0.001, 0.1] |
| 64 | $prefF2MB1$ | Grazing preference fish2 on MB1 | - | 0.25 | [0.001, 0.5] |
| 65 | $prefF2MB2$ | Grazing preference fish2 on MB2 | - | 0.25 | [0.001, 0.5] |

**Table A3.** ECOSMO and ECOSMO E2E model parameters (part 3)



*Author contributions.* HN conceived of and wrote the PSO toolbox and the toolbox application. HN, UD, NB, CS conceived of the manuscript structure. HN wrote the draft, with UD, NB and CS contributing to revisions.

*Competing interests.* The contact author has declared that none of the authors has any competing interests.

395 *Acknowledgements.* The research was funded by I2B-Funds (Innovation-, Information- and Biologisation-Funds)



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
