# Peer review of "Parameterisation toolbox for physical-biogeochemical model compatible with FABM. Case study: the coupled 1D GOTM-ECOSMO E2E for the Sylt-Romo Bight, North Sea"

_EGUsphere, 2024_

## Author Response (AR3)

Thank so much for taking time to read our manuscript and provide valuable suggestions! Yours comments have significantly improved the manuscript.

Below are our responses to your (Reviewer 1, Reviewer 2, and Editor) comments.

**Reviewer 1**

***Specific comments:***

*Abstract*

Line 2: I would argue that several recent studies go beyond a 'trial-and-error' approach. The authors acknowledge that several recent studies have moved beyond a 'trial-and-error' approach. Indeed, relevant studies are cited in the paragraph beginning at line 30. Here, 'trial-and-error' refers to the traditional approach, which is contrasted with more modern methods introduced later in the text. While this approach may still be suitable for smaller problems, the intention was not to disregard recent advancements but rather to provide context for the discussion of newer techniques.

Line 10: Sentence beginning 'The toolbox was able to...' the tool box itself didn't identify the parameters but defined optimal values.
The text was corrected to "The toolbox was able to define optimal values for most of the tuned parameters ...".

*Introduction*

Line 21: are they incomplete or not devised?
The text "and have not been devised" was removed.

Paragraph starting line 30: Yumruktepe et al., 2023 have described a successful parameter optimization framework utilizing ARGO floats and the same (GOTM-FABM-ECOSMO) model set-up as used in this study. The PSO would potentially fit nicely within the ARGO toolbox allowing identification of a parameter set that is appropriate over larger areas without excessive computational requirements.
We are in contact with the author to see if it is possible to apply the PSO to the parameterisation for the model in the paper. We also added it to future work to enhance the robustness of the PSO over extended simulation periods.

Paragraph starting line 46: These hypotheses are referred to again on line 169 but are not mentioned later in the paper, including in the discussion or the section on future work. Has or will the model be used to test these hypotheses?
The application of the PSO parameterised 1D GOTM-ECOSMO E2E for Sylt-Rømø Bight to test mentioned hypotheses was added to the Future work.

*Parameterization toolbox*

Line 60: should bound be boundary?

*Yes, bound was changed to boundary*

Line 64: I find the analogy with the two boats confuses the issue - according to the current description the two boats will quickly converge close to the deepest point between them (but not at the deepest point in the lake). The description beginning on line 77 is easy to follow even for a non-mathematician, so maybe the boat analogy could be removed altogether – if kept it should be improved.
*Yes, the paragraph was removed.*

*Model configuration and set up*

No mention is given to the performance of GOTM in simulating the complex physical regime of the region. This is relevant as poorly constrained physics will impact the final parameter set.

*While we did not explicitly evaluate the GOTM model, we applied relaxation to observed temperature and salinity to ensure that the modeled values closely matched the observations. To clarify this, we have added a figure and a corresponding explanation starting at line 179.*

Line 189 (or 176): Was a one-year spin-up enough to achieve a stable ecosystem state?

*We added sentences to this paragraph to clarify the details of the one-year spin-up.*

The choice of parameter space within which parameters are allowed to vary is not discussed.
*The parameter spaces are provided in Tables A1–A3. They were derived by varying the reference values (also listed in Tables A1–A3) within a range of half to double their original values.*
*We added more details about the parameter space in the paragraph starting at line 211.*

Some model parameters are more tightly constrained than others (based on existing knowledge) so the size of the parameter space within which an individual parameter is allowed to vary may be parameter dependent. Are the authors sure the defined parameter spaces do not exceed meaningful ranges in each individual case?
*We are aware of the problem of the size of the parameter space in the parameterisation problem. Therefore, we rescaled the parameter space to [0,1] in our PSO to avoid the mentioned problem. The parameters were restored to their true range before they enter the biogeochemical model. The description of the parameter space rescaled to [0, 1] is given in the Algorithms section (line 80).*

In cases where the final optimized parameters are significantly different to the reference value some discussion may be merited. For example, looking at Figure 4, the optimal maximum growth rate for large phytoplankton is almost double the reference value. The value of 2.5 is high compared to values typically used and may lie close to the upper boundary of the defined parameter space? Can this high value be justified through consideration of the specific ecosystem processes at work in the study region? Or is

there some compensation within the model (such as high growth rates balance by high grazing rates?)

In general consideration should be given to parameters which exhibit very large changes from the reference values. How does the new (optimal) parameter compare to that used in other models? And does it remain within a scientifically meaningful range? What does this say about the system being studied?

A paragraph starting line 311 was added to address this.

*Results and discussion*

Line 264/5: Why does the model underrepresent phytoplankton biomass at other times of the year?

This part was rewritten to explain the underrepresentation of phytoplankton biomass at other times of the year.

Presumably river input does not account for NO3 removal. Also why does the lack of river input not impact phosphate in the model?

Several sentences were added to answer the question.

The 8-year period simulated is still relatively short and does not indicate robustness over longer decadal/multidecadal periods. The robustness seen may also come at a cost – looking at Figure 5 is there some evidence that the model is tuned towards a mean state and underrepresents interannual variability?

Due to data availability constraints, the longest consistent dataset available for this case study spans an 8-year period. A longer time series would be beneficial for assessing the robustness of our tool. In the future, we may explore another case study with a longer (multi-decadal) dataset.

The explanation for Figure 5, where the model is tuned towards a mean state, is provided in the paragraph beginning at line 353.

*Section 3.2*

297: Have you defined one parameter set, within which several of the parameters can be allowed to vary within a defined range? If you have define several parameter sets which are actually different then the system dynamics cannot be identical. I feel further discussion is needed around this subject. Choosing a parameter set that achieves the right result for the wrong reasons means the parameter set is likely less portable to other locations or time frames and will also likely have implications for the representation of higher trophic levels within the model system.

This issue of multiple parameters sets reproducing observations equally well, despite differences in values, is a fundamental challenge in parameterisation and is often referred to as equifinality. Further discussion of this aspect was given in the paragraph starting line 381.

*Conclusions*

Line 327: Neither the hydrodynamic complexity or the success of GOTM in representing this has been discussed in the paper.
We have added a figure and a corresponding explanation starting at line 179.

Line 333: It may be better to point to the doi given at the end of the paper here? At least an institutional rather than personal site is required to ensure secure long-term access.
The link given here is actually the institutional link of Helmholtz, a research association in Germany. The DOI of zenodo is also given in the code availability section after line 460.

Line 350:
I see the identification of different parameter sets as highlighting the limitation of the model and the method, care should be taken not to achieve the right results for the wrong reasons if the model is to be used to study the system dynamics. Although the possibility of defining stable parameter ranges is, however, a plus.
The paragraph starting at line 381 and 401 were added to discuss on the issue of multiple parameters sets reproducing observations equally well.

Figure A2 and A3: I did not understand why the black dots sometimes appear to be outside the range of the grey dots.
We re-checked the plots again and found a bug in the plotting code that truncated some of the data, causing the black dots to sometimes appear out of range of the grey dots. We have fixed this error and updated the Figures A2 and A3.

*Other minor comments*

The manuscript should be checked for typos/grammatical errors e.g.

Line 1: insert 'an' after 'serve as'. done
Line 2 (or 16?): remove 'in recent years'. done
Line 5 (or 19?): insert a comma after 'system' or rewrite the sentence to be clearer. done
Line 44: remove 'to' after 'also'. done

**Reviewer 2**

***General comments***

1) Although the manuscript is generally clearly written, I found many grammar errors (e.g. number concordance) in the Introduction. Please revise grammar thoroughly throughout the text. Also, pay attention to the formatting of the references.
The manuscript was grammatically checked.

*2) Regarding the PSO algorithm (section 3.1 Demonstration of the PSO toolbox)*

What is the likelihood that the optimization gets trapped in a local minimum? Is there a means to ensure the PSO algorithm is exploratory enough to avoid local minima? Did the authors made tests in this regard? For example, running the PSO against a model-generated dataset where (a) exact parameter values are known, and (b) the optimized model should be able to converge exactly to the solution.
Getting trapped in a local minimum: This might be missing from the manuscript, but in the code of the toolbox, we included perturbation steps. These steps ensure that, at a pre-defined iteration, the parameter is reset to a new value, allowing it to move away from any local minimum it might be drawn to.
We have added text at lines 116 (below the algorithm description), 132, and 226 to clarify the implementation of the perturbation step in our PSO.

I missed here a more extended discussion of how the PSO performs in comparison to other parameter optimization approaches (e.g. in terms of number of iterations, total number of executions needed, convergence speed, convergence accuracy...).
A new paragraph starting line 270 and 285 was added to provide further discussion of how the PSO performs in comparison to other parameter optimization approaches.

*3) Section 3.2* (Multiple parameter sets can reproduce observations equally well despite differences in values) is very welcome, as it puts in the spotlight a common (but often overlooked) problem in parameter optimization. However, it falls short at deciphering whether the similar model outputs with different parameter sets result from mutually-compensating processes or from low sensitivity. I am not proposing authors to discover which is the case, but just to let readers know that different scenarios are possible and discuss them briefly. Again, improved use of the literature will enrich this section.
A new paragraph starting line 381 was added to discuss low sensitivity as one of the causes of equifinality.

*4) Section 3.3* (Optimising model complexity with PSO: top-down control by macrobenthos in the marine ecosystem around Sylt Road). In my modest view this is clearly the most interesting/exciting section. The ability to evaluate models of different complexity in a unified framework is powerful and can illuminate key aspects of ecosystem functioning. Authors may want to further highlight this aspect.
The section was rewritten to highlight the finding.

***Specific comments***

L22: "remain incomplete and have not been devised" please use more appropriate wording. Nobody knows what complete bgc model equations look like.
The text "and have not been devised" was removed.

L27: "alternative" to what? or is parameter optimization just a means to enable simplistic models to reproduce observed state variables and fluxes?
The related sentence has been changed to "Where direct empirical determination is not possible, parameter optimization has been proposed to address these challenges (Fennel et al., 2001; Dowd, 2011). While this approach can help models reproduce observed state variables and fluxes, its broader purpose is to systematically improve model skill by finding parameter sets that optimize the agreement between model outputs and observations."

L32: "subjective"? I guess the authors meant "objective" or at least quantitative, in the sense that a misfit function is minimized. The three papers cited above (Prieß et al., 2013; Falls et al., 2022; Kern et al., 2024) cannot be deemed to be proposing "subjective" approaches.
"subjective" was changed to "objective" as suggested.

L95: perhaps replace neighborhood by population to remain consistent in terms of wording (I understood them as equivalent).
"neighborhood" has been replaced by "population" at this line and other places in the manuscript as suggested.

L133: "A quick guide to the PSO Toolbox:" can possibly be removed, or merged into the title of the subsection (currently "PSO with FABM") to make it more explicit.
"A quick guide to the PSO Toolbox" was deleted and made it a paragraph of section with subtitle "PSO with FABM".

L135: suggest replacing "validated" with the less judgmental "evaluated" (usually preferred in modeling frameworks)
L190: same as L135. Check throughout.
"validated" was replaced by "evaluated" throughout the manuscript as suggested.

L137: "and the intervals and frequency for randomly resetting parameters rather than inferring them from parameters and skill scores from the previous iteration": this is hard to understand, please explain more clearly or give readers more background information.
The text was corrected to "In addition, the users must specify the number of iterations to run the PSO, the number of model evaluations per iteration, and the perturbed iterations where model parameters were reset to avoid getting trapped in a local minimum."

L146: Please explain here what the difference is between ECOSMO and ECOSMO E2E.
L186: see comment on L146.
A new paragraph starting lie 140 was added to provide ECOSMO model description.

L153: it seems "calibration" is used as a synonym for "optimization". Perhaps stick to one term for clarity.
Changed to optimisation.

L159: please specify that the Wadden Sea is at the coastal margin of the North Sea.
The text was added.

L191-193: please refer to Table A1 and distinguish model parameters (lie "aa") using italics, or different typography... to help readers.
It was done as suggested.

L204: is guess "the number of model evaluations per iteration" is equivalent to "population size". Please specify to help readers .
The first occurrence of "the number of model evaluations per iteration" was changed to "population size (i.e., the number of model evaluations per iteration)" and the subsequent occurrences were changed to "population size".

L208: not necessarily the "duration", but the computing time. If all model evaluations are done in parallel for a given iteration step, there should not be noticeable increases in "duration".
"duration" was changed to "computing time". In our implementation, we did not use parallel computing, so there is indeed a significant increase in computing time. However, you raise a good point—parallel computing could significantly speed up the PSO. We will explore how to implement it.

L225: how does this compare to other optimization approaches? e.g., those cited in the Introduction? for example, Falls et al. (2022) saw convergence during the first 10-20 iterations with a biased random key genetic algorithm. Please discuss this aspect more in depth.
A paragraph starting line 270 was added to further discuss on the requested aspect.

L242: in my own experience, convergence speed may be related to sensitivity, with faster convergence for more sensitive parameters. This information might be a useful addition here.
Lines 260 – 268 provide additional information on convergence speed and sensitivity.

L248: "It can be seen that about two thirds of the tuned parameters converge on certain values". And the others? Do they converge? Please rephrase in a less vague way. The meaning of "certain" here is unclear.
The sentence was corrected as "It can be seen that about two thirds of the tuned parameters each converge to a specific values."

Figure 4: why are there several black dots for a given parameter? Do they correspond to different experiments? It feels like some info is missing here.
The caption of Figure 5 (previously Figure 4) was adjusted to show details of the black dots.

L267-L275: it seems that the optimized model better captures the mean seasonal cycle, but still fails to capture the interannual variability…? some explanation, even if speculative, to let readers imagine potential workarounds?
A paragraph starting line 353 was added.

L375: fix repetition in this sentence. Done.

Figure A2 (and elsewhere): how is convergence defined?
The rate of convergence is shown in Appendix Figure A6. When the rate approaches 0 and stays around this 0 value for a long time, it is considered to be converging.

Figures A2-A3: I see that, for some parameters, the best parameter value a at given iteration (black) is far apart from the majority of parameter values evaluated (gray). After several iterations, the black dots and the gray cloud converge. Why does it sometimes take so long? Does this point to the need for adjusting the PSO algorithm to improve performance?
We re-checked the plots again and found a bug in the plotting code that truncated some of the data, causing the black dots to sometimes appear out of range of the grey dots. We have fixed this error and updated the Figures A2 and A3.

**Editor**

Please ask someone to check for minor errors with English/typos – some things still need correcting.

We have carefully checked the manuscript for English language errors and typos, and the corrections have been marked throughout the document.

Title: add 'a' after for

"a" was added.

Abstract:

Line 8: remove 'and' before thus

"and" was removed.

Line 8: rewrite to say 'The effectiveness of the PSO toolbox is….'

It was corrected.

Line 12: 'but resulting in not much difference' should be rewritten, this sounds vague and unscientific.

It was rewritten to: "but leading to only minor variations".

Line 42: are coupled to FABM, not are available in FABM.

It was corrected.

Line 149: rewrite 'a version with a two-group version'

It was rewritten for clarity and marked accordingly in the manuscript.

Line 155: assessed not accessed.

It was corrected.

Paragraph on line 172: No need to list variables or levels you are not using.

The unused variables were removed.

191: liter not litter

It was corrected.

318: I'm not sure why higher growth rates compensate for a lack of transport away from the region. Maybe the opposite should be true?

Thank you very much for pointing this out! That was a misunderstanding on my part. I have reconsidered it and revised the explanation as follows:

"The elevated values of $muPl$ and $muPs$ can be attributed to limitations inherent in both the 1D model and the input data. Phytoplankton growth in the model follows Liebig's law of the minimum (Schrum et al., 2006), where the most limiting factor restricts growth. As the MERRA2 reanalysis data underestimates shortwave radiation (Yingshan et al., 2022), which is used to forcing the model, it is likely to result in lower simulated light levels compared to reality, thereby suppressing phytoplankton growth. To compensate for this reduced growth potential and reproduce the observed phytoplankton biomass, $muPl$ and $muPs$ were parameterised to higher values."

385: 'sloppiness' is not the right word to use here.

It was changed to "The work on sloppiness in the model simulations ….". I kept the word "sloppiness" because it is in the title of the referenced paper.

389-393: I do not think these examples are relevant to your case and they should be removed.

The irrelevant examples were removed.